# Investigation of the Mechanism of Action of *Periploca forrestii* Schltr. Extract on Adjuvant Collagen Rats Based on UPLC-Q-Orbitrap-HRMS Non-Targeted Lipidomics

**DOI:** 10.3390/molecules28196751

**Published:** 2023-09-22

**Authors:** Silan An, Xiaoting Yan, Huaguo Chen, Xin Zhou

**Affiliations:** 1Key Laboratory for Information System of Mountainous Areas and Protection of Ecological Environment, Guizhou Normal University, 116 Baoshan North Rd., Guiyang 550001, China; 15121506267@163.com (S.A.); yanxiaoting9806@sina.com (X.Y.); 2Guizhou Engineering Laboratory for Quality Control & Evaluation Technology of Medicine, Guizhou Normal University, 116 Baoshan North Rd., Guiyang 550001, China; 3The Research Center for Quality Control of Natural Medicine, Guizhou Normal University, 116 Baoshan North Rd., Guiyang 550001, China

**Keywords:** rheumatoid arthritis, *Periploca forrestii* Schltr., non-targeted lipidomics

## Abstract

*Periploca forrestii* Schltr. (*P. forrestii*) is a classical medicinal plant and is commonly used in traditional medicine for the treatment of rheumatoid arthritis, soft tissue injuries, and traumatic injuries. The aim of this study was to evaluate the anti-arthritic effects of three fractions of *P. forrestii* alcoholic extracts (PAE), *P. forrestii* water extracts (PWE), and total flavonoids from *P. forrestii* (PTF) on Freund’s complete adjuvant (FCA)-induced arthritis in rats, and to use a non-targeted lipidomic method to investigate the mechanism of action of the three fractions of *P. forrestii* in the treatment of rheumatoid arthritis. To assess the effectiveness of anti-rheumatoid arthritis, various indicators were measured, including joint swelling, histopathological changes in the joints, serum cytokines (tumor necrosis factor-α (TNF-α), interleukin-1β (IL-1β), interleukin-6 (IL-6)), and the joint inflammatory substance prostaglandin E2 (PGE2). Finally, ultra-performance liquid chromatography–quadrupole-orbitrap–high-resolution mass spectrometry (UPLC-Q-Orbitrap-HRMS) was used to determine the non-targeted lipid histology of the collected rat serum and urine samples to investigate the possible mechanism of action. PWE, PAE, and PTF were all effective in treating FCA-induced rheumatoid arthritis. The administered groups all reduced joint swelling and lowered serum inflammatory factor levels in rats. In the screening of lipid metabolite differences between serum and urine of the rat model group and the normal group, a total of 52 different metabolites were screened, and the levels of lipid metabolites in PWE, PAE, and PTF were significantly higher than those in the normal group after administration. In addition, PWE, PAE, and PTF may have significant therapeutic effects on FCA-induced arthritis by modulating nicotinic acid, nicotinamide, and histidine metabolic pathways.

## 1. Introduction

Rheumatoid arthritis (RA) is a chronic, debilitating autoimmune disease characterized by synovial hyperplasia, increased inflammatory cell numbers, angiogenesis, joint destruction, and loss of function. RA can lead to complications such as cardiovascular disease, testicular inflammation, and interstitial lung disease [1]. Currently, drugs used to treat RA include analgesics, non-steroidal anti-inflammatory drugs, and disease-modifying anti-rheumatic drugs, among others [2]. Although these drugs have some efficacy, long-term use is not only expensive, but also has more obvious side effects. Natural medicines are widely recognized for their high efficacy and low toxicity [3,4]. Existing studies have shown that natural plant extracts have therapeutic or alleviating effects on RA. For example, Guo’s experimental results showed that the water extract of *Polygonum orientale* L. can improve RA, while Cheng et al. found that ethanol extract of Deer velvet can relieve RA symptoms [5,6]. Therefore, searching for therapeutic drugs for RA from plants is feasible and has great potential.

*Periploca forrestii* Schltr. (*P. forrestii*) is a traditional medicinal plant distributed in southwestern China, and is included in the Chinese Materia Medica (Miao Medicine Volume) as a Miao medicine. The name, morphology, origin, and efficacy of the medicinal parts of *P. forrestii* have been comprehensively recorded in the Dictionary of Traditional Chinese Medicine, Guizhou Herbal Medicine, and Yunnan Herbal Medicine [7]. *P. forrestii* contains cardiac glycosides, quinones, flavonoids, phenylpropanoids, triterpenoids, steroids, volatile oils and ceramides, and other chemical constituents [8]. Pharmacological studies have shown that *P. forrestii* has various biological activities, including inhibition of acetylcholinesterase, anti-inflammatory, antioxidant, analgesic, anti-tumor, and immune regulation [7,9]. In traditional medicine, *P. forrestii* is commonly used to treat various diseases such as soft tissue injuries, RA, and gouty arthritis [9].

Currently, there is a large amount of pharmacodynamic evidence for the treatment of RA with *P. forrestii*. For example, the ethanol extract of *P. forrestii* extracted by Dong et al. was able to alleviate ear edema and foot edema in rats with xylene and carrageenan models [10]. Liu et al.’s study found that the saponin constituents of *P. forrestii* could significantly improve joint swelling, inhibit articular bone erosion, and reduce the levels of IL-6 and TGF-β1 in splenocytes of rats in the adjuvant arthritis model [11].

In addition, our group has also carried out a large number of studies related to the pharmacodynamic evaluation of *P. forrestii* for the treatment of RA in the previous period. For example, we established a Freund’s complete adjuvant (FCA)-induced arthritis model in rats, and used rheumatoid bone pain capsule (RBPC) and aspirin enteric-coated tablets as positive control drugs, respectively, and conducted pharmacodynamic evaluations on different doses of water (0.2190, 0.1095, and 0.0547 g/kg) and ethanol extracts of *P. forrestii* (0.0168, 0.0084, and 0.0042 g/kg) were evaluated pharmacodynamically. The results showed that both water and alcoholic extracts of *P. forrestii* were effective in reducing the inflammation of adjuvant-type arthritis in rats. X-ray imaging and histopathological sections of rats also showed a more significant improvement in the *P. forrestii* administered group compared with the model group. This suggests that *P. forrestii* has significant anti-RA activity and has the potential to become an excellent therapeutic agent for RA. However, there is a lack of in-depth mechanistic studies on *P. forrestii* for the treatment of rheumatoid arthritis. Therefore, in-depth mechanisms of action studies are necessary for its development and utilization.

Lipids play fundamental roles in a variety of physiological and pathophysiological situations, as constituents of biomembranes, sources of energy, and signaling molecules. Lipidomics uses analytical chemistry to purposefully analyze the lipidome of organisms, which can effectively analyze changes in lipid levels and lipid metabolic networks in different physiological and pathological states, thus effectively identifying new signaling molecules, revealing potential mechanisms of different physiological and pathological conditions, discovering potential lipid biomarkers for early diagnosis and prognosis of diseases, screening drug targets and evaluating drug efficacy, and opening up new ideas for disease prevention and treatment [12,13].

On this basis, the three best active extracts, *P. forrestii* water extract (PWE), *P. forrestii* alcoholic extract (PAE), and *P. forrestii* total flavonoids (PTF), were selected in this study to investigate the mechanism of anti-RA action of Bupleurum officinale.In the present study, we firstly established a rat RA model induced by Freund’s complete adjuvant (FCA) and re-evaluated the efficacy of the PWE, PAE, and PTF were re-evaluated for their efficacy. At the same time, a non-targeted lipidomic approach was adopted to screen the serum and urine samples of rats for lipid metabolism differentials, and the lipid metabolism pathways were explored in conjunction with the relevant databases, with the aim of elucidating the mechanism of action of *P. forrestii* against RA from the perspective of lipid metabolism.

## 2. Results

### 2.1. Results of Pharmacodynamic Studies

#### 2.1.1. Effect of *P. forrestii* in Adjuvant-Induced Arthritis (AA) Rats

##### Effect of *P. forrestii* on Body Weight, Joint Swelling in AA Rats

The pathogenetic features of Freund’s complete adjuvant (FCA)-induced animal model were similar to those of RA mainly manifested in two periods: primary arthritis and secondary arthritis. Primary arthritis began to swell a few hours after the injection of FCA, and then subsided 3–5 d later, while secondary sexual joints mainly manifested symmetrical arthralgia, which generally began to swell 8–9 d after the first injection of FCA [14]. After FCA was injected subcutaneously into the left hind toe of rats, fever, swelling, and foot licking by rats appeared in the joints of the left hind limb within 6 h, indicating successful modeling. As can be seen in Figure 1, the FCA-induced joint tissues of rats showed significant swelling compared with the Normal group, indicating successful modeling.

Table 1, Table 2 and Table 3 show the swelling and body weight of the left and right joints of the rats, respectively. As can be seen from Table 1, the left joint of the model rats showed significant swelling compared to the normal group (*p* < 0.05, 0.01), and after 6 d of modeling, the swelling of the joints of the rats in the PTF group was significantly reduced compared to the model group (*p* < 0.05). As can be seen in Table 2, after the 9 d of modeling, the symmetrical joint (right joint) of the rats showed swelling, which was alleviated in PTF, PWE, and PAE groups compared to the diameter of the joints of the rats in the model group, and statistically different (*p* < 0.05) compared to the model group. The results in Table 3 showed that the body weight of rats in the AA group decreased significantly from 16 to 23 days (*p* < 0.05, 0.01), whereas the body weight of rats in the PWE, PAE, and PTF groups all increased significantly from 20 to 24 d compared to the AA group (*p* < 0.05, 0.01).

The role of cytokines in the pathogenesis of RA is the result of their involvement in several immune pathways. IL-6, IL-1β, PGE2, and TNF-α, pro-inflammatory cytokines produced mainly by macrophages and monocytes, are critical to the pathogenesis of RA [15,16,17,18]. The levels of inflammatory cytokines in rats with FCA-induced RA are shown in Figure 2a–d.

Compared with the normal group, the levels of IL-6, IL-1β, PGE2, and TNF-α in the AA model group were significantly (*p* < 0.05, 0.01) increased. However, compared with the AA group, rats in the PWE, PAE, and PTF groups exhibited significantly decreased (*p* < 0.05, 0.01) levels of IL-6, IL-1β, PGE2, and TNF-α.

##### Effect of *P. forrestii* on Histopathological Examination of Joints in AA Rats

RA features include synovial hyperplasia, increased numbers of inflammatory cells, angiogenesis, joint destruction, and loss of function, resulting in, particularly, inflammation of the hands, wrists, and symmetrical foot joint [19]. Histological evaluation of left ankle joint sections in rats was performed on the 19th day after treatment. Cartilage histopathological sections in rats were stained with HE. The results are shown in Figure 3.

Rats in the normal group showed no pathological symptoms related to arthritis. In contrast, histological sections of the AA model rats showed significant pathological changes, including uneven articular surfaces and connective tissue infiltration, cartilage cell proliferation, synovial thickening, and a small amount of inflammatory cell infiltration. The results are shown in Figure 3. Compared with the AA group, joints from the PWE, PAE, and PTF groups showed a significantly reduced inflammatory response, with a reduction in hyperplasia, inflammatory cell infiltration, and cartilage surface erosion.

### 2.2. Non-Targeted Lipidomics Data Analysis

#### 2.2.1. Multivariate Analysis of the Serum and Urine Profiles for Model Establishment

Untargeted lipidomics is a very effective method for discovering lipid biomarkers in biological samples, not only lipids that can be used as diagnostic biomarkers, but also those that are effective therapeutic or potential drug targets for diseases [20]. In order to study the metabolic differences and trends in the six groups of rats, principal component analysis (PCA) was established to distinguish the Normal, AA, RBPC, PWE, PAE, and PTF groups. The PCA scores of the six groups of rats are shown in Figure 4. In Figure 4, all PCA models consisted of six components, urine ESI+ R2X (cum) = 0.536 (Figure 4a) and ESI− of R2X (cum) = 0.595 (Figure 4b), and serum ESI+ R2X (cum) = 0.531 (Figure 4c) and ESI− R2X (cum) = 0.608 (Figure 4d), both in positive-ion or negative-ion mode, there was a significant group separation between the normal and AA groups, and PWE, PAE, PTF and RBPC were all between the AA and normal groups. The results showed that there was a significant difference in lipid metabolites between the AA and normal groups, and the overall lipid metabolism level tended to move toward the normal group when the AA rats were given *P. forrestii* extract, suggesting that lipid metabolism disorders induced by RA can be ameliorated by the treatment with *P. forrestii* extract.

In order to further validate the difference between normal and AA groups to screen for differential lipid metabolites, this study performed an OPLS-DA modeling analysis of samples between normal and AA groups, as shown in Figure 5 [21]. Taking the OPLA-DA between the two groups of normal and AA groups as an example, in the serum positive-ion model of Figure 5a—serum+, R2X(cum) = 0.567, Q2(cum) = 0.887, and in the serum negative-ion model of Figure 5a—serum-, R2X(cum) = 0.714, Q2(cum) = 0.814, and the differences between the serum positive-ion model and the negative-ion model of R2X and Q2 were both less than 0.4 and Q2 was greater than 50%, indicating a good fit and predictive ability between the models. In the urine positive-ion model of Figure 5a—urine+, R2X(cum) = 0.763, Q2(cum) = 0.563, and in the urine negative-ion model of Figure 5a—urine−, R2X(cum) = 0.775, Q2(cum) = 0.538, and the difference between R2X and Q2 was less than 0.4, and Q2 was greater than 50% for both the urine positive and negative-ion models, suggesting a good fit and prediction ability between the models. This indicates a good fit and predictive ability between the models.

In addition, in order to prevent the overfitting of the OPLS-DA model and also to screen the lipid metabolites with significant differences between the normal and AA groups, the data under the OPLS-DA model were validated by 200 substitutions, and as shown in Figure 5b, the Y-intercepts of Q2 for the positive- and negative-ion modes of serum and urine were all less than 0, which indicated that the model was not overfitted. In summary, the results of the OPLS-DA model and the response ordering permutation validation showed that the constructed model had good statistical significance, and there was a good separation between normal and AA in the OPLS-DA model, indicating that there were lipid metabolism differences between the two groups.

#### 2.2.2. Identification of Potential Biomarkers

Screening and structural analysis of differential metabolites are the focus and difficulty of current non-targeted lipidomics research. In this study, the screening conditions for differential lipid metabolites were as follows: FC > 2.0 or < 0.5, VIP > 1.0, and *p* < 0.05 were used as the screening conditions for significant differential lipid metabolites between normal control and model groups. In general, lipids with VIP value > 1 were considered significant, lipids with FC value > 2.0 or < 0.5 were considered significantly up-regulated or down-regulated, and a *p*-value of 0.05 (*p* < 0.05) indicated a significant difference. After cross-screening and structural characterization, a total of 52 significantly different lipid metabolites were screened out in rat serum and urine. As shown in Table 4, these 52 significantly different lipid metabolites were 23 fatty acyl groups, 4 phospholipids, 6 glycerophospholipids, 1 sterolipid, and 18 polyketides. The lipid structure analysis method used in this study was an accurate search of LIPID MAPS and related literature by raw mass spectrometry information. The total ion flow diagrams of rat serum and urine in positive and negative ion mode are shown in Appendix A. Detailed details of the lipids (e.g., *m*/*z*, retention time, molecular weight, etc.) are shown in Appendix A.

#### 2.2.3. Evaluation of the Treatment Effect of *P. forrestii* on AA Rat

Clustering heat map analysis was performed for 52 differential lipid metabolites in each group of normal, AA, RBPC, PTF, PWE, and PAE (Figure 6). As shown in Figure 6, red color indicates lipid up-regulation, and blue color indicates lipid down-regulation. Among the 52 differential lipid metabolites, 17 differential lipid metabolites were significantly up-regulated in AA rats compared with the normal group (*p* < 0.05), including three fatty acids, two sphingolipids, three glycerophospholipids, one sterolipid, and eight polypolyethylenes. There were 35 differential lipid metabolites significantly down-regulated, including 20 fatty acids, 2 sphingolipids, 3 glycerophospholipids, and 10 polypolyethylenes. The above results indicated that the relative contents of lipid metabolites in normal and AA rats had a large difference, while the levels of lipid metabolites in AA rats significantly converged to those of the normal control group after administration of PWE, PAE, and PTF, suggesting that the *P. forrestii* extract had a modulating effect on RA lipid metabolism disorders. 

#### 2.2.4. Lipid Metabolic Pathway Analysis

The 52 metabolites showing up-regulated or down-regulated expression were imported to MetPA for metabolic pathway analysis. This analysis indicated that the nicotinamide metabolic pathway and the histidine metabolic pathway were particularly perturbed (Figure 7). These results suggest that *P. forrestii* may reduce the inflammatory response by influencing the nicotinamide and histidine metabolic pathways.

Further analysis of lipid pathways using the KEGG database (http://www.kegg.jp/, accessed on: 25 May 2023) confirmed these results. The impact of the nicotinate and nicotinamide metabolism pathway was 0.13816, and that of the histidine metabolism pathway was 0.22131. The KEGG database (http://www.kegg.jp/, accessed on: 25 May 2023) and related literature were further searched, and a map of RA-related lipid metabolic pathways was compiled, as shown in Figure 8.

## 3. Discussion

Rheumatoid arthritis (RA) is a systemic autoimmune disease with a high disability rate, which can cause systemic organ system damage [19,22]. Currently, the treatment of RA is mainly based on remission, and the commonly used drugs include non-steroidal anti-inflammatory drugs, targeted biologics, targeted small-molecule drugs, etc. [23]. However, both single-drug treatment and a combination of multiple drugs can be used. However, it is difficult to avoid the serious side effects of either single-drug therapy or a combination of multiple drugs [24]. Traditional Chinese medicine (TCM), which treats diseases from multi-targets, multi-levels, and holistic regulation with low toxicity and side effects, has natural advantages in the treatment of RA, and it has become a research hotspot to find anti-RA drugs from TCM [25]. As one of the common herbs used by Miao people in Guizhou Province, *P. forrestii* has been developed and utilized as clinical drugs, such as Hei Gu Teng Chasing the Wind and Activating the Lines Capsule, Hei Gu Teng Chasing the Wind Liquid, and so on [25]. Although clinical trials have been conducted to prove its effectiveness in the treatment of RA, its lack of more in-depth research on the mechanism of action has prevented its larger-scale promotion.

In the present study, an untargeted lipidomic approach was used to investigate the mechanism of action of Bupleurum officinale in the treatment of RA rats. Serum and urine were collected from normal and model rats for lipidomic analysis, a total of 52 significantly different lipid metabolites were identified, and metabolic pathway enrichment analysis revealed that these 52 different lipid metabolites were mainly enriched in the nicotinic acid and nicotinamide metabolism and histidine metabolism, but it was not clear how the extract of *P. forrestii* regulated these two pathways. We hypothesized that PWE, PAE, and PTF might regulate the niacin and nicotinamide metabolic pathway and histidine metabolic pathway from two aspects.

It has been found that in active RA, high concentrations of M1 macrophages are present in the synovium, and activated macrophages can exacerbate RA by producing pro-inflammatory cytokines and chemokines, such as interferon-gamma (IFN-γ) and tumor necrosis factor-alpha (TNF-α) [26], which lead to the destruction of articular cartilage and subchondral bone [27]. The results of this study showed that all three administration groups of PWE, PAE, and PTF could reduce the serum TNF-α level and decrease the PEG2 content in the joint leachate [28]. Therefore, we hypothesized that B. nigra might reduce the serum levels of TNF-α, IL-1β, and IL-6 and block the NF-κB signaling pathway in RA rats by modulating the nicotinic acid and nicotinamide metabolism pathway and histidine metabolism pathway, thus alleviating the RA symptoms [29].

On the other hand, after oral administration to RA rats, the extract was not fully degraded and absorbed into the digestive system, but was transported to the intestines via the gastrointestinal tract, where it was broken down and utilized by the intestinal flora. First, it was found that the intestines of RA patients were rich in Prevotella, while the number of Bacteroides fragilis was low [30]. Pseudomonas fragilis is important for Treg function, and high levels of copri and similar species are associated with low levels of beneficial microorganisms that are thought to modulate the immune system [31]. *P. forrestii* extract may be regulating lipid metabolism by affecting the structure of both of these flora, thereby modulating the species of gut flora and further affecting the gut-liver axis metabolism [32]. Secondly, the extract may become food for the microorganisms in the intestinal tract and be broken down into small molecules such as short-chain fatty acids, and these metabolites may be broken down into small molecules such as vitamins under certain conditions [33], which may be absorbed by the intestinal mucosa into the bloodstream and transported to other organs, thus regulating lipid metabolism, and thus affecting the niacin and nicotinamide metabolic pathways as well as the histidine metabolic pathway.

In conclusion, *P. forrestii* is a natural Chinese medicine that has now been proven to have good therapeutic efficacy for RA in some studies, but its mechanism of action is still unclear. Although the present study elucidated part of the mechanism of action of black bone vine in the treatment of RA from the perspective of lipidomics, it still lacks more in-depth mechanistic mechanism studies, such as the specific pharmacokinetic study of *P. forrestii*. Finally, there are still unresolved issues in this study, such as the aqueous and alcoholic extracts of *P. forrestii* are not highly purified compounds, and their therapeutic effects may be a combination of multiple compounds. The extraction temperature of the lipids may also have an effect on the experimental results. If the samples are kept at room temperature for too long, the enzymes and chemical degradation processes may lead to lipid peroxidation or hydrolysis, so attention is paid to the extraction time and temperature when extracting the lipids [34].

## 4. Materials and Methods

### 4.1. Reagents and Materials

Freund’s complete adjuvant (FCA) was purchased from Sigma Aldrich Chemie Gmbh (St. Louis, MA, USA). All the enzyme-linked immunosorbent assay (ELISA) kits (IL-1β, IL-6, PGE2, and TNF-α) were purchased from Shanghai Mlbio Biotechnology Co., Ltd. (Shanghai, China). Rutin was obtained from Guizhou Dida Technology Co., Ltd. (Guiyang, China); the purity of these reagents was higher than 98% as determined by HPLC. Rheumatoid bone pain capsules (Batch No. 180304) were purchased from Sinopharm Group Jingfang (Anhui) Pharmaceutical Co., Ltd (Xuancheng, China). Acetonitrile and methanol of HPLC grade were obtained from Tedia Co., Inc. (Fairfield, OH, USA). Ultrapure water was used for all solutions and dilutions (ultrapure water was prepared by the subject TM-D24UV ultrapure water machine). Other chemicals and solvents were of analytical or HPLC grade.

### 4.2. Sample Preparation

#### 4.2.1. *P. forrestii* Sample Preparation

*P. forrestii* was collected in Guiyang (Guizhou Province, China), and was identified by Huaguo Chen, Professor at Guizhou Normal University. *P. forrestii* was pulverized in a pulverizer, passed through an 80-mesh sieve, and then placed in a ziplock bag and stored in a dry, ventilated place at 10 °C for later use. Preparation of PAE: Take 200 g of *P. forrestii* powder, extract with 80% ethanol 1000 mL reflux 3 times, each time 2 h, filtration, filtrate concentrated under reduced pressure, and then vacuum drying, to get PAE extract. Preparation of PWE: Take 200 g of *P. forrestii* powder, extract with 2000 mL of distilled water at 80 °C 3 times, each time for 2 h, filtration, filtrate concentrated under reduced pressure, and then dry in vacuum to obtain PWE extract. PTF prepared in the laboratory at >90% [35]. According to the group’s previous study, the optimal doses of PWE, PAE, and PTF for the treatment of rheumatoid arthritis were all determined to be 0.675 g/kg/d.

#### 4.2.2. Positive Drug Preparation

The dosage of the rheumatoid bone pain capsule was converted to the rat daily dose (0.21 g/kg/d) and mixed with distilled water to form a suspension. Dosage refers to rheumatoid bone pain capsule instructions: 2 capsules 3 times daily, 0.3 per capsule. 

### 4.3. Establishment of Adjuvant-Induced Arthritis (AA) Rat and Treatment Regimen

#### 4.3.1. Animals

SD rats (180 ± 200 g body weight), specific pathogen-free grade (SPF), Certificate (No. SCXK 2014-0011), were obtained from Changsha Tianqin Bio-Technology Co. Ltd. (Changsha, China). Half of the rats were male, and half were female. Rats were housed in an environmentally controlled breeding room (24 ± 1 °C and natural light–dark cycle) for a week, with libitum access to water and rodent chow before starting the study. All experiments were carried out between 09:00 and 17:00 h. 

#### 4.3.2. Model Building of Adjuvant-Induced Arthritis (AA) Rat and Experimental Grouping

Sixty rats were randomly divided into 6 groups of 10 rats each, male and female, and grouped as follows: blank control group (Normal), model control group (adjuvant-induced arthritis (AA)), positive control group (rheumatoid bone pain capsule (RBPC)), *P. forrestii* water extraction (PWE) group, *P. forrestii* alcoholic extract (PAE) group and *P. forrestii* total flavonoids (PTF) group. The RBPC group was given rheumatic bone pain capsules, and PAE, PWE, and PTF were given relative to each other. Normal and AA were given equal amounts of saline, the route of administration was gavage administration once a day, the volume of administration was 10 mL/kg/d in all cases, the administration was started 3 days after modeling, and the administration was continued for 21 days.

The rats in each experimental group and the model group were injected with 0.1 mL of FCA subcutaneously into the left toe of the rats, and the rats in the blank control group were injected with an equal amount of saline.

All experimental procedures were performed according to the National Guide on the Proper Care and Use of Animals in Laboratory Research. The study was also approved by the Animal Ethics Committee.

### 4.4. Pharmacodynamic Study

#### 4.4.1. Evaluation of Arthritis

The diameters (mm) of the left ankle joints of rats in each group were measured by a person with a vernier caliper at 10:00 a.m. every day before and 6 h, 1 d, 2 d, 3 d, 4 d, 5 d, 6 d, and 7 d after modeling. On the 8th day after modeling, the diameter (mm) of the right ankle joint of rats in each group was measured every other day. The body weight of rats in each group was recorded every 4 days to observe the effect of drugs on the body weight of rats with adjuvant arthritis.

#### 4.4.2. Collection of Serum, Urine

At the end of the experimental period, rats were fasted for 12 h after administration of the last dose. Animals were lightly anesthetized with ether, urine samples collected, and blood taken from the femoral artery, after which the rats were sacrificed. Blood samples were stored at 25 °C for 30 min, followed by centrifugation at 3000 rpm for 5 min at 4 °C to obtain serum. The serum and urine of animals were stored at −80 °C before extraction of lipids.

#### 4.4.3. Preparation of Rat Serum Samples, Determination of Serum Cytokine Levels and PEG2, a Joint Inflammatory Substance

After anesthesia, blood was taken from the femoral artery of rats, and the blood was collected in clean and sterile 1.5 mL centrifuge tubes and then centrifuged (4 °C, 3000 rpm, 10 min). After 30 min of resting at room temperature, serum was collected and stored in a refrigerator at −80 °C for spare use. The blood was thawed at 4 °C, and was tested in accordance with the methods of the instructions of the IL-1β, IL-6, and TNF-α kits, respectively.

In the left posterior joints of rats in each group, the skin of the joints was removed and put into test tubes containing 5 mL of saline, left for 12 h, centrifuged, the supernatant was taken, and its OD value was determined in strict accordance with the instructions of the rat PEG2 kit.

#### 4.4.4. Histopathological Assessment of Joints

After rapid removal of the right ankle and surrounding skin, the joints were fixed in 4% paraformaldehyde and decalcified in 10% ethylene diamine tetraacetic acid (EDTA) for up to 25 days at 4 °C. After decalcification, the tissues were dehydrated, processed, and then embedded in paraffin. Serial paraffin sections (5 μm) were stained with hematoxylin and eosin (HE) and examined under a light microscope (400× magnification) (American Molecular Devices, Model Max Plus 384) [36,37]. The severity of arthritis was assessed based on pathologic changes such as inflammatory cell infiltration, synovial hyperplasia, joint swelling and inflammation, hemangiomas, and cartilage and bone destruction [38,39].

### 4.5. Non-Targeted Lipidomics Analysis

#### 4.5.1. Preparation and Extraction of Lipid Samples

Serum and urine samples stored at −80 °C were thawed at room temperature. A 150 μL volume of the sample was added to 1.5 mL EP tubes to which was added 600 μL of ice-cold methanol, 150 μL of pure water, and 450 μL of dichloromethane. The mixtures were shaken for 30 s, and a further 150 μL of water was added and vortexed for 1 min. Samples were centrifuged at 12,000 rpm for 15 min at −6 °C; the lower liquid was transferred to a 1.5 mL EP tube and dried under nitrogen. To the residue, 200 μL of a mixture of acetonitrile, isopropanol, and water in the ratio of acetonitrile/isopropanol/water (65/30/5 V:V:V) was added and then centrifuged for 5 min at 4 °C and 8000 rpm, and the supernatant was stored at −80 °C for UPLC-Q-Orbitrap-HRMS analysis [40].

#### 4.5.2. Chromatography and Mass Spectrometry

For lipidomics experiments, a Thermo Scientific Quadrupole-Orbitrap-high resolution Mass Spectrometry coupled to a Dionex UltiMate 3000 system was used, with an ESI ionization source in both positive- and negative-ion modes. All samples were maintained at 4 °C in the autosampler prior to injection (5 μL) onto the column in duplicate. Lipids were separated on a Phenomenon ACE Excel 1.7 C18-AR UPLC PR column (100 mm × 2.1 mm) at a mobile phase flow rate of 0.3 mL/min, and the temperature of the column was maintained at 40 °C. Mobile phase A was acetonitrile containing 0.1% acetic acid, and mobile phase B was 0.1% acetic acid in water. The gradient conditions were as follows: 0–5 min, 30% A; 5–30 min 30–90% A; 35–40 min 30% A. The mass spectrum parameters of the positive and negative model were as follows: capillary temperature and probe heater temperature to 320 °C, AGC target to MS 1 e^6^ to MS/MS 2 e^5^, sheath gas to 35 arb, and AUX gas to 10 arb. The positive-ion and negative-ion modes were set to 3.5 and 3.2 kV as well as an acquisition rate of 5 spectra/s. Mass spectrometric data were acquired in a full-scan mode (*m*/*z* 100–1000).

#### 4.5.3. Multivariate Data Analysis and Biomarkers Identification

The Xcalibur software (Version 2.2, Thermo Scientific, Waltham, MA, USA) and Progenesis QI (Nonlinear Dynamics) were used for data acquisition and pretreatment. The data pretreatment process included the baseline filtering and calibration of the baseline, peak alignment, peak identification, and integration of the peak area. After any known pseudo-positive peaks, such as peaks caused by noise and column bleed, were removed from the data set, the CSV files were obtained. The resulting data matrix which consists of sample code, peak names, retention time, and peak intensities, was further processed using Microsoft Excel 2013. Total chromatographic area normalization was applied to reduce the deviation between each sample.

Multivariable statistical analysis, including orthogonal projection to latent structures–discriminant analysis (OPLS-DA) and principal component analysis (PCA), were performed by SIMCA-P (Version 14.1, Umetrics, Umea, Sweden). The data were log10-transformed with centralized processing, the model was validated by SIMCA-P internal cross-validation, and the resulting variable importance projection (VIP) values were integrated with the data [41,42]. The OPLS-DA illustrates the differences between normal and model groups by investigating those lipids that show the largest differences in the data, and the larger the VIP value, the more likelihood of the lipid being a significant contributor. The differential lipids between the model and normal groups were selected based on the VIP scores in OPLS-DA (VIP > 1), Student’s *t*-test *p*-value (*p* < 0.05), and fold change (FC, FC > 2.0 or <0.5).

The identification of lipids was based on the accurate mass and MS/MS fragmentation spectra acquired in the automatic, data-dependent MS/MS analysis during the initial data acquisition. Matched LipidMaps (http://www.lipidmaps.org, accessed on: 25 May 2023), the Human Metabolome Database (HMDB) (http://www.hmdb.ca/, accessed on: 25 May 2023), and published fragmentation patterns were used to structurally identify lipids.

#### 4.5.4. Pathway Analysis of Lipid Differentials

Pathway analysis based on the identified lipids was carried out using MetPA 4.0 (http://www.metaboanalyst.ca/, accessed on: 25 May 2023) (Impact >  0). The Kyoto Encyclopedia of Genes and Genomes (KEGG) pathway database (www.genome.jp/kegg/, accessed on: 25 May 2023) was used for biological interpretation. The rat (Rattus norvegicus) pathway library was selected for identifying possible metabolic pathways.

### 4.6. Statistical Analysis

The data are presented as means ± standard error of the mean (SEM). Differences between different groups were compared with Student’s *t*-test, with *p* < 0.05 set as the level of statistical significance. SIMCA-P (Version 14.1, Umetrics) is used for multivariate statistical analysis.

## 5. Conclusions

In the present study, an RA animal model was established by injection with FCA. As shown by serum biochemical indices and analysis of weight change, joint swelling, cytokine levels, and histopathology, symptoms of inflammation were clearly alleviated in the drug-treated groups. The results show that the Miao medicine *P. forrestii* has a significant therapeutic effect on RA caused by FCA. UPLC-Q-Orbitrap-HRMS non-targeted lipidomics was used to investigate the mechanism of action of *P. forrestii* to treat RA. A total of 52 differential lipids were screened and identified in the serum and urine of the model group and normal group, and these lipids were submitted to online lipid databases to identify possible metabolic pathways involved. Two pathways were identified: the nicotinate and nicotinamide, and the histidine metabolic pathways. Significant changes in the up and down-regulation of various lipids were observed in the drug-treated groups, with analysis indicating that PAE was the most effective of the extracts. These results provide insight into the mechanisms of the Miao medicinal *P. forrestii* remedy RA and offer a reference for future investigations into the therapeutic applications of Miao medicinal remedies.

## Figures and Tables

**Figure 1 molecules-28-06751-f001:**
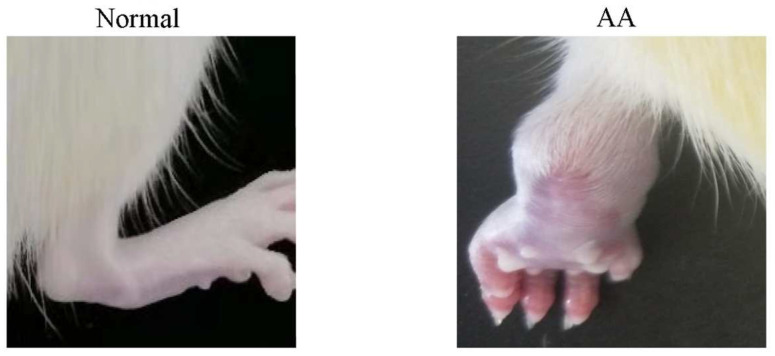
Joints of normal and model (AA) rats.

**Figure 2 molecules-28-06751-f002:**
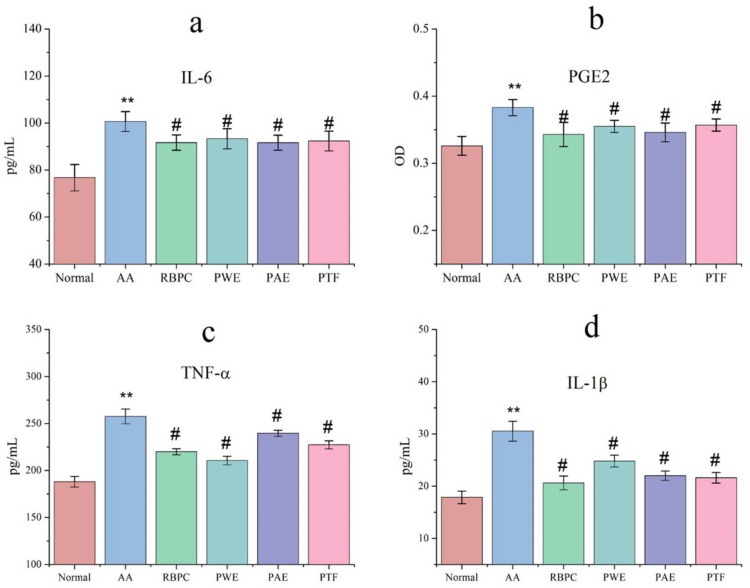
Effects of *P. forrestii* IL-6 (pg/mL) (**a**), PGE2 (OD) (**b**), TNF-α (pg/mL) (**c**), IL-1β (pg/mL) (**d**) levels in FCA-induced arthritic rats. Data are expressed as mean ± SD (*n* = 8). ^#^
*p* < 0.05 vs. Normal group, ** *p* < 0.01 vs. AA group.

**Figure 3 molecules-28-06751-f003:**
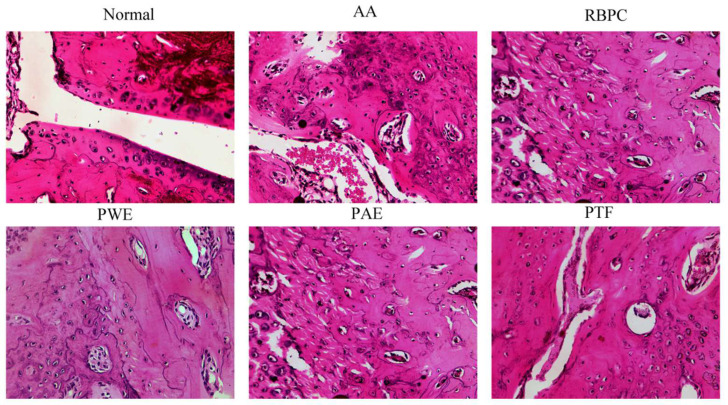
Effect of *P. forrestii* on histopathological change of ankle joint cartilage in AA rats. Representative pictures are shown. (Hematoxylin eosin stained × 400).

**Figure 4 molecules-28-06751-f004:**
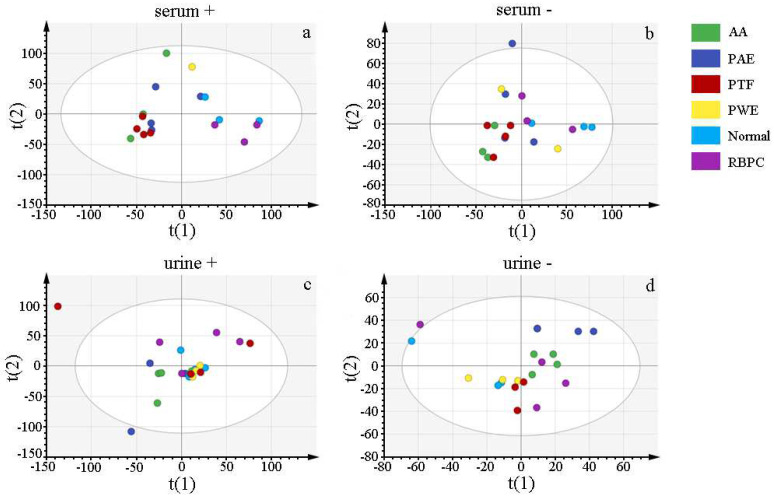
PCA score plot of ESI^+^ and ESI^−^ of serum and urine of six components. ((**a**) serum positive ion pattern, (**b**) serum negative ion pattern, (**c**) urine positive ion pattern, (**d**) urine negative ion pattern.)

**Figure 5 molecules-28-06751-f005:**
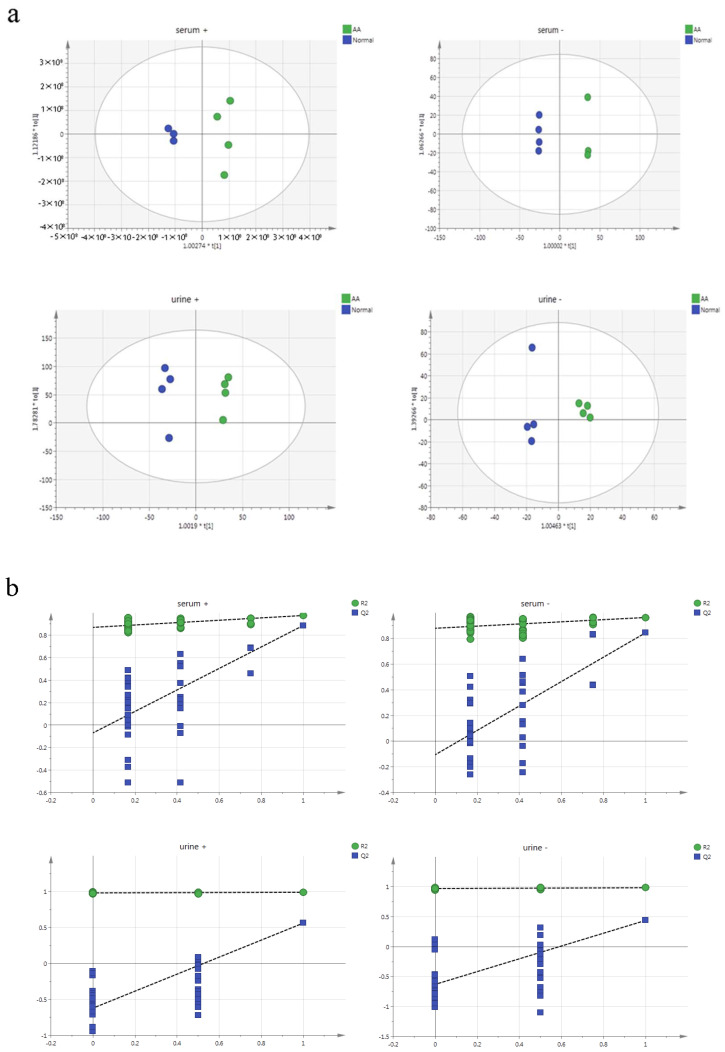
OPLS-DA score plots (**a**) and corresponding permutation test (200 times) (**b**) obtained from UPLC-Q-Orbitrap-HRMS, which includes serum and urine under positive ions and negative ions. R2: the degree of fit of the response model, Q2: the predictive power of the response model.

**Figure 6 molecules-28-06751-f006:**
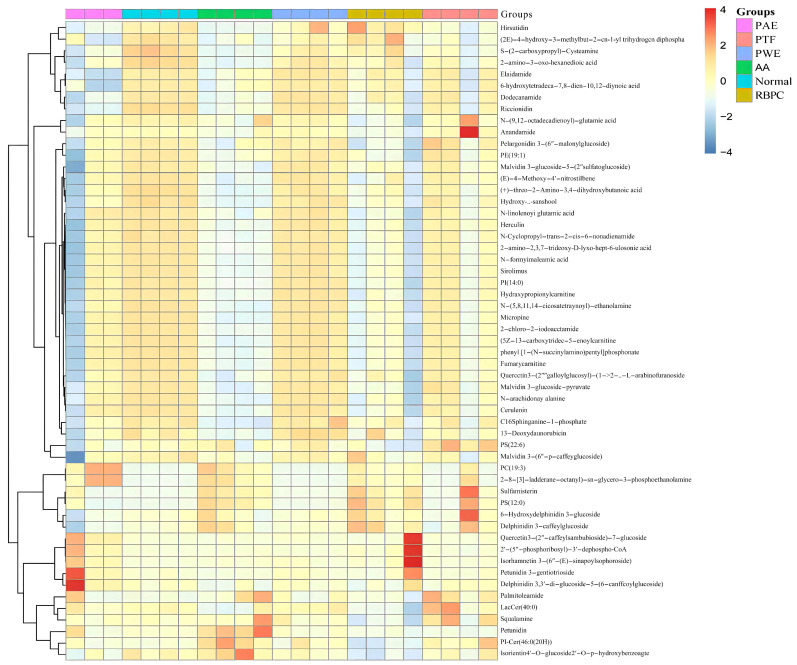
Heat map of the most abundant lipid metabolites in rat serum and urine (showing changes in levels of 52 lipids in response to the normal, PWE, PAE, and PTF treatment groups compared to the AA group).

**Figure 7 molecules-28-06751-f007:**
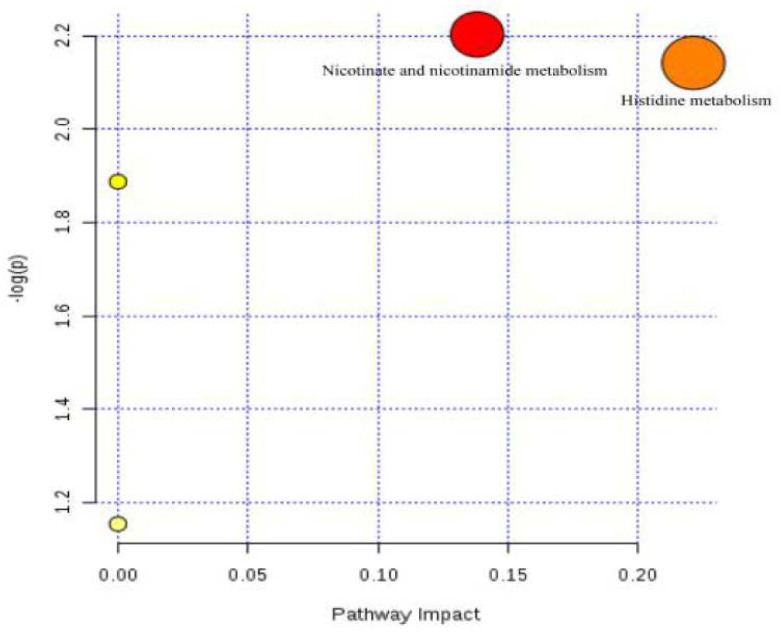
Disturbed lipid metabolic pathways of RA rat as visualized by bubble plots. The color and size of each circle represents the *p*-value and the pathway influence factor (darker the color, larger the *p*-value; larger the bubble, larger the influence factor).

**Figure 8 molecules-28-06751-f008:**
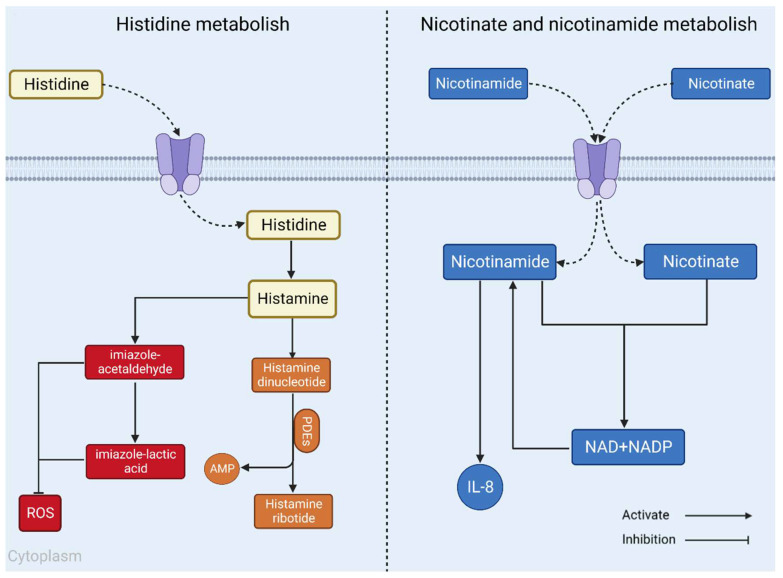
Map of RA-related lipid metabolism pathways.

**Table 1 molecules-28-06751-t001:** Effect of *P. forrestii* extraction on primary adjuvant arthritis in rats, means ± SD (*n* = 8).

Group	0 h	6 h	24 h	2 d	3 d	4 d	5 d	6 d	7 d
Normal	6.19 ± 0.26	6.29 ± 0.2 *	6.24 ± 0.11 *	6.22 ± 0.17 *	5.93 ± 0.16 *	6.26 ± 0.31 *	6.17 ± 0.12 *	6.20 ± 0.11 *	6.09 ± 0.11 *
AA	6.20 ± 0.14	8.44 ± 0.47	8.69 ± 0.53	8.61 ± 0.64	8.37 ± 0.59	8.33 ± 0.43	8.39 ± 0.46	8.42 ± 0.48	8.33 ± 0.56
RBPC	6.24 ± 0.16	8.24 ± 0.72	8.67 ± 0.52	8.61 ± 0.51	8.59 ± 0.52	8.58 ± 0.43	8.50 ± 0.45	8.29 ± 0.36	8.10 ± 0.45
PWE	6.27 ± 0.09	8.44 ± 0.45	8.50 ± 0.53	8.12 ± 0.39	6.61 ± 0.32 ^#^	8.47 ± 0.38	8.47 ± 0.42	8.30 ± 0.48	8.20 ± 0.47
PAE	6.20 ± 0.26	8.46 ± 0.47	8.19 ± 0.46	8.13 ± 0.38	8.33 ± 0.39	8.24 ± 0.59	8.26 ± 0.51	8.18 ± 0.61	8.07 ± 0.61
PTF	6.22 ± 0.13	8.25 ± 0.54	8.28 ± 0.57	8.24 ± 0.51	8.33 ± 0.29	8.30 ± 0.31	8.12 ± 0.37	7.09 ± 0.47 ^#^	6.59 ± 0.45 ^#^

* *p* < 0.05, Model vs. Normal group, one-way ANOVA and Dunnett’s multiple comparisons t. ^#^
*p* < 0.05, Drug administration group vs. Model group, one-way ANOVA and Dunnett’s multiple comparisons t.

**Table 2 molecules-28-06751-t002:** Diameter of the right ankle joint of rats with FCA-induced arthritis (mm), means ± SD (*n* = 8).

Group	d 9	d 11	d 13	d 15	d 17	d 19	d 21	d 23
Normal	6.19 ± 0.11 *	5.90 ± 0.12 *	6.16 ± 0.11 *	6.18 ± 0.14 *	6.07 ± 0.09 *	6.13 ± 0.05 *	6.10 ± 0.06 *	6.05 ± 0.08 *
AA	6.70 ± 0.16	6.69 ± 0.21	6.73 ± 0.20	6.73 ± 0.14	6.75 ± 0.10	6.69 ± 0.06	6.72 ± 0.08	6.71 ± 0.05
RBPC	6.50 ± 0.19	6.51 ± 0.26	6.63 ± 0.15	6.63 ± 0.10	6.48 ± 0.12	6.48 ± 0.08	6.45 ± 0.08	6.43 ± 0.12
PWE	6.48 ± 0.20	6.44 ± 0.22	6.52 ± 0.13	6.57 ± 0.11	6.39 ± 0.14 ^#^	6.45 ± 0.06	6.48 ± 0.04	6.46 ± 0.08
PAE	6.50 ± 0.14	6.21 ± 0.15 ^#^	6.65 ± 0.09	6.63 ± 0.06	6.58 ± 0.08	6.56 ± 0.07	6.26 ± 0.04 ^#^	6.47 ± 0.06
PTF	6.29 ± 0.18 ^#^	6.26 ± 0.17 ^#^	6.38 ± 0.08 ^#^	6.40 ± 0.11 ^#^	6.30 ± 0.05 ^#^	6.46 ± 0.07	6.24 ± 0.06 ^#^	6.24 ± 0.08 ^#^

* *p* < 0.05, Model vs. Normal group, one-way ANOVA and Dunnett’s multiple comparisons t. ^#^
*p* < 0.05, Drug administration group vs. Model group, one-way ANOVA and Dunnett’s multiple comparisons t.

**Table 3 molecules-28-06751-t003:** Body weight of rats with FCA-induced arthritis (g), means ± SD (*n* = 8).

Group	Modeling	d 2	d 5	d 9	d 13	d 16	d 20	d 24
Normal	215.55 ± 0.87	222.47 ± 0.65	228.78 ± 0.32	232.11 ± 1.06	235.19 ± 0.96	241.11 ± 0.53 *	247.68 ± 0.86 **	251.90 ± 0.72 **
AA	216.68 ± 0.82	219.76 ± 0.96	220.21 ± 1.02	223.71 ± 0.89	226.46 ± 1.32	221.46 ± 0.96	213.84 ± 0.56	201.02 ± 0.31
RBPC	217.68 ± 0.85	217.09 ± 0.98	220.79 ± 0.54	228.40 ± 0.68	231.19 ± 1.21	238.18 ± 0.36	241.55 ± 0.78 ^#^	243.20 ± 0.69 ^#^
PWE	214.36 ± 0.63	216.67 ± 0.85	220.19 ± 0.64	228.62 ± 0.89	230.59 ± 1.24	234.87 ± 1.33	236.79 ± 1.45 ^#^	240.90 ± 0.65 ^#^
PAE	215.90 ± 1.32	219.88 ± 0.98	222.39 ± 0.89	225.99 ± 1.01	230.41 ± 1.32	232.81 ± 0.78	236.17 ± 0.65 ^#^	239.80 ± 0.58 ^#^
PTF	213.97 ± 1.01	219.24 ± 0.65	220.54 ± 0.87	222.40 ± 0.85	230.93 ± 0.87	235.37 ± 0.89	237.58 ± 2.01 ^#^	242.50 ± 0.78 ^##^

* *p* < 0.05, ** *p* < 0.01, vs. Normal group, one-way ANOVA and Dunnett’s multiple comparison t. ^#^
*p* < 0.05, ^##^
*p* < 0.01, vs. Model group (AA), one-way ANOVA and Dunnett’s multiple comparison t.

**Table 4 molecules-28-06751-t004:** Differential serum and urine metabolites (total of 52).

Class	Lipid	MW	*m*/*z*	RT (Time)	FC	VIP	*p*-Value
Fatty acyls (FA)	N-Cyclopropyl-trans-2-cis-6-nonadienamide	193.147	192.938	1.05	3.361	1.493	**
	Herculin	251.225	250.903	1.05	5.044	1.445	**
	Anandamide	323.282	323.222	10.19	5.510	1.417	**
	Cerulenin	223.121	223.028	16.34	3.193	1.474	**
	2-chloro-2-iodoacetamide	218.891	218.931	1.03	5.025	1.548	**
	N-arachidonoyl alanine	375.277	375.193	1.93	5.859	1.428	**
	N-(9,12-octadecadienoyl)-glutamic acid	409.283	409.210	13.09	7.823	1.353	**
	N-linolenoyl glutamic acid	407.267	407.238	14.58	4.937	1.361	**
	Dodecanamide	199.194	199.024	26.28	10.825	1.611	**
	6-hydroxytetradeca-7,8-dien-10,12-diynoic acid	232.110	232.022	29.61	0.412	1.652	**
	N-(5,8,11,14-eicosatetraynoyl)-ethanolamine	339.219	339.120	13.92	0.243	1.612	**
	(5Z)-13-carboxytridec-5-enoylcarnitine	399.262	399.196	14.31	0.253	1.614	**
	2′-(5″-phosphoribosyl)-3′-dephospho-CoA	899.157	899.210	18.11	8.556	1.657	**
	Hydroxy-α-sanshool	263.188	263.188	2.00	0.035	1.632	**
	Fumarycarnitine	259.106	259.049	3.33	1.668	3.613	**
	N-formylmaleamic acid	143.002	142.995	14.30	5.843	1.683	**
	Palmitoleamide	253.241	253.217	15.14	2.789	1.487	**
	Hydroxypropionylcarnitine	233.126	233.036	1.62	3.20	1.631	**
	2-amino-2,3,7-trideoxy-D-lyxo-hept-6-ulosonic acid	191.079	191.106	4.90	4.284	1.641	**
	S-(2-carboxypropyl)-Cysteamine	163.066	163.039	27.01	0.261	1.722	**
	(+)-threo-2-Amino-3,4-dihydroxybutanoic acid	135.053	135.044	27.80	0.316	1.741	**
	phenyl [1-(N-succinylamino) pentyl]phosphonate	343.118	343.167	11.6	1.391	9.689	**
	Elaidamide	281.272	280.9828	21.77	0.444	0.647	**
	2-amino-3-oxo-hexanedioic acid	175.048	175.039	28.47	0.399	1.758	**
	C16Sphinganine-1-phosphate	353.233	353.156	9.62	0.217	1.628	**
Sphingolipids (SP)	Micropine	265.204	265.252	13.79	0.244	1.631	**
	Sulfamisterin	461.206	461.224	13.48	3.346	1.347	**
	PI-Cer(46:0(2OH))	981.724	981.649	20.93	14.138	1.671	**
	LacCer(40:0)	947.727	947.637	15.51	7.456	1.411	**
Glycerophospholipids (GP)	PC(19:3)	531.332	531.364	16.74	19.931	1.356	**
	2-(8-[3]-ladderane-octanyl)-sn-glycero-3-phosphoethanolamine	487.306	487.339	16.60	10.786	1.399	**
	PS(12:0)	455.192	455.124	13.50	2.808	1.299	**
	PS(22:6)	879.505	879.409	19.39	23.384	1.527	**
	PE(19:1)	477.322	477.391	22.11	8.005	1.659	**
	PI(14:0)	544.265	544.272	15.44	4.174	1.680	**
Polyketides (PK)	(E)-4-Methoxy-4′-nitrostilbene	255.089	255.099	2.05	0.030	1.635	**
	Malvidin 3-(6″-p-caffeyglucoside)	655.166	655.069	13.47	4.752	1.444	**
	Petunidin 3-gentiotrioside	803.225	803.196	15.13	10.825	1.610	**
	Delphinidin 3,3′-di-glucoside-5 -(6-caffeoylglucoside)	951.241	951.234	15.91	24.67	1.236	**
	6-Hydroxydelphinidin 3-glucoside	481.098	481.133	13.48	4.068	1.291	**
	13-Deoxydaunorubicin	513.199	513.104	13.50	3.209	1.428	**
	Delphinidin 3-caffeylglucoside	627.135	627.050	13.52	3.714	1.349	**
	Pelargonidin 3-(6″-malonylglucoside)	519.114	519.149	15.14	20.063	1.537	**
	Quercetin3-(2‴-caffeylsambubioside)-7-glucoside	920.222	920.250	17.09	14.725	1.551	**
	Petunidin	317.066	317.065	26.24	3.515	1.655	**
	Malvidin 3-glucoside-pyruvate	561.124	561.181	15.92	13.305	1.577	**
	Isorientin4’-O-glucoside2″-O-p-hydroxybenzoagte	730.174	730.251	17.13	2.868	1.419	**
	Quercetin3-(2‴-galloylglucosyl)-(1->2)-α-L-arabinofuranoside	748.149	748.246	14.36	2.676	1.491	**
	Hirsutidin	345.097	345.097	15.26	0.260	1.671	**
	Malvidin3-glucoside-5-(2″-sulfatoglucoside)	735.144	735.071	14.09	6.989	1.386	**
	Riccionidin A	285.04	285.0493	27.25	0.319	1.645	**
	Isorhamnetin 3-(6″-(E)-sinapoylsophoroside)	846.222	846.2310	16.53	11.717	1.659	**
	Sirolimus	913.555	913.582	20.91	27.309	1.526	**
Isoprenoids (PR)	(2E)-4-hydroxy-3-methylbut-2-en-1-yl trihydrogen diphosphate	262.0007	261.9093	29.99	0.485	1.669	**
Sterol Lipids (ST)	Squalamine	627.464	627.386	13.47	5.311	1.470	**

** represents *p*-value of <0.05, which is one of the screening conditions for significantly different lipid metabolites between the normal and model groups.

## Data Availability

Data is contained within this article and Appendix A.

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
