# Peer review of "Investigation of the Mechanism of Action of Periploca forrestii Schltr. Extract on Adjuvant Collagen Rats Based on UPLC-Q-Orbitrap-HRMS Non-Targeted Lipidomics"

_molecules, 2023, doi:10.3390/molecules28196751_

Round 1
Reviewer 1 Report
Review:
General Comments:
- Author list seems not complete?
- Same accounts for the subscripts…only affiliations from 1-3 and the * are indicated. However # is not?
- Why is the corresponding author not on the last spot?
- Please be very careful with annotated and identified….this makes a great difference
- Please refer to the lipid identification guidelines
Steps Toward Minimal Reporting Standards for Lipidomics Mass Spectrometry in Biomedical Research Publications - PMC (nih.gov)
Recommendations for good practice in MS-based lipidomics - ScienceDirect
- It is not clear, what the impact of the study is. What do the results suggest for future research and potential therapeutic treatment?
Abstract:
- Please double check for introduced acronyms
- Line 26: Is this really histology? Please rephrase and use a more fitting wording
Introduction:
- Line 43 – references are missing
- Sentence until line 74 – please check grammar (one verb too much)
- Line 86-88: I find the sentences very strong – please reduce the
- Line 103: The way the sentence is worded, it is not clear whether the three “best parts” (please rephrase this too) are all in the water extract…Please rephrase this sentence and the corresponding sentence in the abstract.
Results:
- Line 121/122: Do you have pictures to support the numbers of the swelling? I suggest to insert them either here in the main manuscript or in the supplementary part.
- Line 123-133: this part seems a bit confusing. Can you restructure it?
- The table heading needs to be placed over the Table 1
- Table 2 heading: Here, it is not clear what you mean with secondary. Please explain this to the reader better
- Table 3: Please rephrase the header (“Weight in rats”?)
- Also, please throughout improve the tables and their information provided as well in the headers and footers
- Please use different colours in Fig 1
- Please provide the complete list of all lipids annotated (with details such as m/z, molecular weight, ret times MS1 and MS2 etc) as supplementary material
- Please improve the quality of Figure 3, I cannot read the legend
- Can you combine the information provided in 4 PCA plots in 1 plot?
- Why does Figure 4 have a “b” in front?
- Section 2.2.2: Please improve the language. Line 235: What do you mean with fragmentation pathways?
- Why do you show these 52 lipids? Not clear
- Were only these 52 “biomarker” lipids used to create the PCAs? If not, please still provide a full list of all compounds included.
- 52 biomarker lipids seems a lot. Are these really all biomarkers?
- What about the samples, that presented as “outliers” in the heat map?
- 2.2.4; How many lipids of these two pathways were identified prior? I.e. how many of the lipids belong to the histidine metabolism? How many to the nicotinate and nicotinamide metabolism?
- It is not completely clear what was compared here…healthy vs RA? Please be more precise here. Depending on which data you used, your statement and conclusion might have to be rephrased.
Discussion:
- Line 325: Did you perform the extraction at RT? Please add this information in the materials and methods part. Please also state in the discussion how this can be improved,
- Line 333: Please be very careful with the wording “identification”. There are guidelines for lipid identification. Did you follow the guidelines?
Materials and Methods:
- Please be more precise and exact throughout the materials and method section. It should allow the reader to re-do the experiment. This would not be possible in the current state.
- Line 346: For metabolic studies, it is common practice to use not deionized but HPLC grade water. Did you analyse a water blank?
- Line 354: What does “cool” mean? Please be more precise and use exact temperatures
- Line 356: What does concentrated mean? How much was it concentrated? To dryness?
- Line 375: Do you mean the ethics or the experiments? Do you have an ethics approval number?
- Line 435: Was the dried residue added to the acetonitrile, isoporanol, water or the other way around?
- Line 439: Which orbitrap system was used? Please give the exact name and model. Also, a quadrupole-orbitrap system appears strange…please double check
- Was an ESI or HESI source installed?
- Line 445: What is eluent A?
- Line 451: in discovery mode? Did you already have a list of the lipids?
- Were only compounds with MS2 included in the annotation process?
- Please be careful with identification and annotation and correct throughout the manuscript
- Did you use an internal standard?
Please improve the English throughout the manuscript. Especially in section 2.2.2 for instance.
Overall formatting is very poor.
Reviewer 2 Report
The article is written in a good language but some minor mistakes are there throughout the manuscript which needs to be rectified. Some questions are there which need to be answered:
Q.1 Formatting needs to be done throughout the manuscript?
Q2:Some typo errors are there which needs to be rectified?
Q3: Revise the language throughout the manuscript
The article is written in a good language but some minor mistakes are there throughout the manuscript which needs to be rectified. Some questions are there which need to be answered:
Q.1 Formatting needs to be done throughout the manuscript?
Q2:Some typo errors are there which needs to be rectified?
Q3: Revise the language throughout the manuscript
Reviewer 3 Report
Manuscript ID: molecules-2576958
The paper entitled “Investigation of the mechanism of action of Periploca forrestii Schltr. extract on adjuvant collagen rats based on UPLC-Q-Orbitrap-HRMS non-targeted lipidomics”. The authors reported the mechanism of anti-rheumatoid arthritis action of three fractions of Periploca forrestii on Freund's complete adjuvant (FCA)-induced arthritis in rats. Moreover, the authors reported a non-targeted lipidomic approach to investigate the mechanism of action of P. forrestii in the treatment of rheumatoid arthritis. This is an interesting paper, and manuscript is well organized. However, for publication a minor revision is needed.
I have just few comments in order to improve the final version of the manuscript before publication, please find them below:
Line 5: The part of the names of the authors must be revised.
Line 47: the scientific name should be in italic
The introduction is too long (lines 68-89 it is better to put this part in the discussion)
Table 1: You must put the title on a separate line
Figure 3: Low resolution
Line 237: Sphingolipids
Reviewer 4 Report
Dear editor
1. The manuscript explores the therapeutic potential of Periploca forrestii Schltr. (P. forrestii) in treating rheumatoid arthritis (RA) and the need for in-depth mechanistic studies to understand its mode of action. The authors provide a comprehensive background on RA, the limitations of current treatments, and the potential of natural medicines. They highlight the pharmacological properties of P. forrestii and its historical use in traditional medicine. The study incorporates previous research conducted by the authors and emphasizes the significance of lipidomics in investigating the mechanisms underlying P. forrestii's anti-RA activity.
The authors offer a comprehensive introduction to RA, its impact, and the limitations of existing treatments, creating a clear context for their study. The rationale for exploring natural medicines and the potential of P. forrestii is well-established
Materials and Methods:
The Materials and Methods section of the manuscript provides a detailed description of the reagents, materials, and sample preparation methods used in the study. The section outlines the sources of reagents, kits, and chemicals, ensuring transparency and reproducibility. Sample preparation procedures for P. forrestii are well-documented, including details on collection, identification, extraction ratios, and purification methods. The optimal doses for treatment are clearly stated, adding to the methodological clarity.
